# A New 3D Printing System of Poly(3,4-ethylenedioxythiophene) for Realizing a High Electrical Conductivity and Fine Processing Resolution

**DOI:** 10.3390/mi11121120

**Published:** 2020-12-17

**Authors:** Katsumi Yamada, Junji Sone

**Affiliations:** 1Department of Life Science and Sustainable Chemistry, Tokyo Polytechnic University, 1583, Iiyama, Atsugi, Kanagawa 243-0297, Japan; 2Department of Applied Computer Science, Tokyo Polytechnic University, 1583, Iiyama, Atsugi, Kanagawa 243-0297, Japan; sone@cs.t-kougei.ac.jp

**Keywords:** micro-nano 3D printing, poly(3,4-ethylenedioxythiophene), electrical conductivity, processing resolution

## Abstract

Micro-nano 3D printing of the conductive 3,4-ethylenedioxythiophene polymer (PEDOT) was performed in this study. An oil immersion objective lens was introduced into the 3D photofabrication system using a femtosecond pulsed laser as the light source. As a result, the processing resolution in the horizontal and vertical directions was improved in comparison to our previous study. A relatively high electrical conductivity (3500 S/cm) was found from the obtained 3D PEDOT micro-structures. It is noteworthy that the high conductivity of the PEDOT was obtained in the mixed state with an insulating Nafion sheet.

## 1. Introduction

Maruo et al. were the first to obtain 3D micro-structures in 1997, by the two-photon excitation of a ultraviolet (UV) curable resin [1]. Multi-photon excitation can occur in an extremely narrow space, and is suitable for micro-nano-scaled 3D photofabrication by finely scanning the focus of a femtosecond pulse laser in the X-Y-Z dimensions. However, the 3D structures of the ultraviolet (UV) curable resin obtained by this method do not have a function reflecting their size, and are just like sculptures. Therefore, although it was a successful demonstration of a new technology, it was necessary to give the 3D micro-structures a useful function in order to become a popular technology.

For this technology, Wegener et al. focused on photonic crystal and a metamaterial as the attractive target. Since they achieved a processing resolution of several hundred nm by improving the 3D photofabrication conditions of the UV curable resin, they could obtain some photonic crystals in the near infrared region [2]. The obtained 3D structures of the UV curable resin were used as a gold plating template to realize a 3D metal metamaterial [3]. Furthermore, they succeeded in improving the processing resolution of the 3D structures to several tens of nm by using stimulated emission depletion (STED) technology [4]. Previously, the processing resolution in the vertical direction of the 3D structures was not very high, but it was dramatically improved by employing the STED technology. However, many processes were required for the method of using a UV curable resin as a template to obtain 3D structures consisting of metallic materials. Blasco et al. employed a mixture of a UV curable resin and metal precursors as the photofabrication material. After the photofabrication, the formed 3D structures were chemically reduced in order to obtain metal in the 3D structures. They found that the 3D structures had a relatively high electrical conductivity [5].

The conductive polymers have some attractive characteristics, such as that they can switch between a high electrical conductivity like metallic materials, and can insulate materials by about six orders of magnitude. If the optical metamaterial is composed of these materials, the refractive index and the resonance wavelength can be switched by energy stimulation (such as an electric signal or light illumination). However, these materials are basically insoluble in most solvents, and have a poor processability. This is probably the reason why these materials are not used in more commercial fields. The authors have studied some two-dimensional images of conducting polymers using photo-induced electron transfer (one-photon process) between a ruthenium complex (as the photo-sensitizer) and methylviologen. The authors started studying the photo-polymerization of conducting polymers and micro-nano 3D photofabrication with multi-photon excitation of the photo-sensitizer in 2006 [6,7,8,9].

Anomalous transmission of light at a specific wavelength was found from a plate with pin holes coated with a heavy-doped polypyrrole by Matsui et al. in 2006 [10]. This study was expected to open the way for the active tuning of metamaterials. The coexistence of suitable processing methods for conducting polymers, and a high electrical conductivity, is required to realize switchable metamaterials.

In recent years, the authors have found a relatively high electrical conductivity during the study of the micro-nano 3D photofabrication of polypyrrole [11]. The electrical conductivity value was 410 S/cm in the doped state. It is worth noting that this value was not found for pure polypyrrole, but in the mixed state inside the Nafion sheet, which conducts little electricity in the dry state.

Poly(3,4-ethylendioxythiophene) (PEDOT) is one of the most industrially successful conductive polymers, and is characterized by having an excellent high electrical conductivity and high transparency in the conductive state [12,13,14].

The micro-nano 3D photofabrication of PEDOT, and the evaluation of its electrical conductivity were carried out using a new system in the study.

## 2. Materials and Methods

### 2.1. Materials

All chemicals were purchased from Sigma-Aldrich (St. Louis, MO, USA) and used as received. A propylene carbonate solution was used as the polymerization solution, and contained 0.1 M tetrabutylammonium tetrafluoroborate (TBABF_4_), 25 mM dimer of 3,4-ethylendioxythiophene (bis-EDOT, Azuma Co., Ltd., Ichihara, Chiba, Japan), 1 mM methylviologen (MV^2+^), and 1 mM tris(2,2′-bipyridyl)ruthenium complex [Ru(bpy)_3_^2+^]. Since the EDOT monomer, having a higher oxidation potential, was hard to polymerize by the Ru(bpy)_3_^2+^, the bis-EDOT, having a lower oxidation potential, was employed as the start material of the polymerization. The chemical structure of the bis-EDOT is shown in Figure 1. A 10 mm × 10 mm piece of Nafion 212 sheet (50 μm thickness, Sigma-Aldrich) was used as the deposition support of the PEDOT. The sheet was immersed in the polymerization solution, then illuminated to polymerize the bis-EDOT.

### 2.2. Photo-Fabrication

The excitation of Ru(bpy)_3_^2+^ was provided by a mode-locked Ti/sapphire laser (λ = 850 nm, Tsumami, Spectra-Physics, Inc., Mountain View, CA, USA). The original laser pulse, with a repetition rate of 80 MHz, was reduced to 8 MHz by a pulse selector (Spectra-Physics) with an acousto-optic modulator. The laser beam was tightly focused by an oil immersion objective lens (x60, N.A. = 1.42, PLAPON60XO, Olympus Corporation, Shinjuku, Tokyo, Japan). The illuminated areas (laser focal point) were transferred under computer-control by shuttering the beam and driving the substrate using an X-Y-Z piezo stage (MB-140XYZLS, MESS-TEK Co., Ltd., Wako, Saitama, Japan). In this 3D printing system, the scanning conditions of the laser focal point, such as pitch (distance to the next point) and wait (illumination time at the point), were explored in order to form the line deposition.

### 2.3. Evaluation of the Electrical Conductivity

After the photo-fabrication, the Nafion sheet was immersed for 2 h in an acetonitrile (Sigma-Aldrich) solution containing 1.0 M FeClO_4_ (Sigma-Aldrich) to dope ClO_4_^−^ into the PEDOT deposition. The Nafion sheet was then washed and dried under vacuum conditions. The washing was carried out with acetonitrile as solvent. In order to arrange the electrodes at both ends of the PEDOT line deposition, Au thin films were immobilized on both sides of the Nafion sheet after the doping process. Their electrical conductivities were measured with a source meter (Model 2400 Series Source Meter, Keithley Instruments, Inc., Cleveland, OH, USA) at 298 K under vacuum conditions by application of DC voltages in the full-doped state.

## 3. Results and Discussion

### 3.1. Processing Resolution

The authors have already succeeded in the micro-nano 3D photofabrication of PEDOT [15]. In a previous study, a dry type objective lens was used for the photofabrication system. The laser light emitted from the objective lens was introduced to the Nafion sheet through the air, the slide glass, and the polymerization solution. Since the group velocity dispersion and out of focus were induced by several interfaces with different refractive indexes, the processing resolution of the obtained PEDOT structures would be lower. Micrographs for the obtained line pattern of the PEDOT from the dry objective are shown in Figure 2. The surface image was observed from the surface of the Nafion sheet with a microscope, and the cross-sectional image was observed by cutting the sheet into small pieces and exposing the cross section. In particular, the cross-section of the horizontally drawn line patterns was quite distorted, and the aspect ratio, which is the ratio of the size in the vertical direction to the line width, was around 7.5 (repetition rate of 8 MHz).

Therefore, the reduction of the interface was planned by immersing the oil immersion objective lens into the polymerization solution. As a result of the photofabrication with the oil immersion objective (repetition rate of 8 MHz), micrographs for the obtained line pattern of the PEDOT are shown in Figure 3. After enlarging these micrographs of the obtained PEDOT line depositions, the line width and the height measurements were carried out using an electronic digital vernier caliper (CD-15CPX, Mitutoyo, Kawasaki, Kanagawa, Japan). The relationships between the laser exposure and the line width are shown in Figure 4a. The line widths linearly increased with the increasing incident laser exposure, shown in Figure 4a. The minimum line width of 0.69 μm was obtained at the exposure value of 0.185 mWs, and was better than the diffraction limit (0.73 μm) of the optical system. The relationships between the laser exposure and the aspect ratio are shown in Figure 4b. Compared to the results of the photofabrication with the dry objective (Figure 2), the aspect ratios were significantly smaller than two. According to these results, clear improvements in the processing resolution and the reproducibility were realized by the new method. The reason is considered to be that the adoption of the oil immersion objective is effective in generating the multi-photon excitation of the ruthenium complex.

An interesting phenomenon was observed during the 3D photofabrication by PEDOT. When the conductive polymer has an absorption band in the wavelength region (around 850 nm) of the laser light source, the laser light does not reach the lower positions of the previously formed deposition, and thus the polymerization will not occur at that position. However, PEDOT has a high transmittance in these wavelength ranges in the conductive state, and the laser light can reach the lower position of the previously formed deposition. Therefore, a flat spiral structure extending in the vertical direction of the Nafion sheet was designed as a good demonstration. The design of the photofabrication and the cross-sectional micrograph image of the Nafion sheet after the photofabrication are shown in Figure 5. As in Figure 5a, the laser focus was scanned inside the Nafion sheet and from outside the spiral. The result with a dry objective lens (repetition rate of 8MHz) is shown in Figure 5b, and although the design shape could be reproduced, it is unclear. On the other hand, the result with an oil immersion objective lens (repetition rate of 8 MHz) is shown in Figure 5c, and the spiral shape was more clearly reproduced than in Figure 5b by the dry objective lens. This reflected the improvement of the aspect ratio due to the adoption of the oil immersion objective lens.

### 3.2. Electrical Conductivity

In order to evaluate the electrical conductivity of the PEDOT micro-structure obtained by this 3D photofabrication system, line patterns were designed that linearly penetrated the Nafion sheet in the thickness direction. After the photofabrication, two gold electrodes were immobilized on both sides of the obtained Nafion sheet by sputtering to form a sandwich-type device. DC voltages were applied to these electrodes, and the current values were measured. The conductivity measurements with polypyrrole often failed in our previous study. The cause of this problem was considered as follows. When the photofabrication was performed for the conductivity evaluation, the laser focus was scanned in the vertical direction, 10 μm from under the outside (from the downstream of the optical path) of the Nafion sheet, and reached the upper outside. (Focus scanning 1 as shown in Figure 6) At this time, if the polypyrrole was not exposed on the surface of the Nafion sheet, the contact between the electrode and the polypyrrole could not be made, and the measurement was not successfully completed. In particular, the contact is probably hard to form under the outside surface of the Nafion sheet. If the exposure time on the position is short, multi-photon excitation can occur immediately, but the subsequent polymerization process occurs slowly because the monomer supply is a slow diffusion-controlled reaction. It is considered that the position of the polymer deposition will be slightly shifted in the inside of the Nafion sheet surface. Therefore, the scanning method of the laser focus was changed based on this hypothesis. As the first step, the laser focus was scanned once from the lower outside into the inside of the Nafion sheet. As the second step, the laser focus was reversed in the opposite direction, and scanned until it went out of the Nafion sheet. As the final step, the laser focus was scanned in the opposite direction again, and the scanning was performed from the opposite Nafion sheet surface to the outside (Focus scanning 2 as shown in Figure 6). A cross-sectional microscopic image of the Nafion sheet as a result of the photofabrication of PEDOT by the focus scanning 2 is shown in Figure 7. Some penetrations of the PEDOT linear patterns were observed in the Nafion sheet. From the left, photofabrication was performed with different incident laser exposures of 0.015, 0.019, 0.033, and 0.041 mWs (repetition rate of 8 MHz), and PEDOT linear patterns with different line widths could be confirmed. As the laser scanning was turned back many times near the lower surface position, the line width of the resulting PEDOT line patterns were slightly thicker at that position. For evaluation of the electrical conductivity, the PEDOT line pattern was formed in the Nafion sheet by the incident laser exposure of 0.041 mWs. (The line width of the product pattern was 3 μm.) The obtained current–voltage characteristics are shown in Figure 8. A linear relationship between the applied voltage and current is observed in the Figure 8. When the applied voltage was 0.1 V, the current was 5 mA. The electrical conductivity obtained as the reciprocal of the volume resistivity was about 3500 S/cm in the doped condition. In recent years, the electrical conductivity of PEDOT has been improved and increased from several hundred S/cm to about 20 times higher, and the highest value reached was 8797 S/cm for single crystal nanowires [16]. Our value of 3500 S/cm obtained even in the mixed state inside the Nafion sheet is remarkable among the other 3D printed conducting polymers [17,18,19]. There have been some reports that higher conductivities were realized from conductive polymer nano-tubes or nano-wires [16,20]. It is expected that these polymer chains are oriented in the nano-structures. In our system, the line widths are several microns, thus it is considered that this relatively high conductivity is due to another mechanism. One of them is the alignment of the polymer chains due to an extremely strong electrical field generated by focusing the femtosecond pulse laser [21]. At present, we are studying the influence of the laser conditions and focus scanning methods on the electrical conductivity of the resulting PEDOT 3D structure.

## 4. Conclusions

The processing resolution of the PEDOT photofabrication was improved with the introduction of the oil immersion objective lens. A relatively high electrical conductivity of 3500 S/cm was obtained from the PEDOT 3D structures. This value is surprising, and was even obtained in a mixed state. In the future, a switchable metamaterial must be realized by the fine 3D structure of PEDOT obtained from this system.

## Figures and Tables

**Figure 1 micromachines-11-01120-f001:**
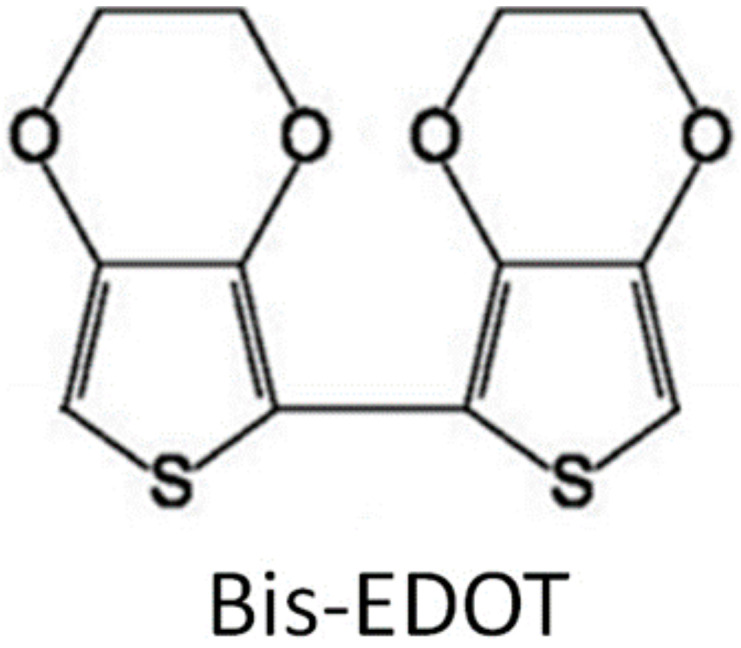
Chemical structure of dimer of 3,4-ethylendioxythiophene (bis-EDOT).

**Figure 2 micromachines-11-01120-f002:**
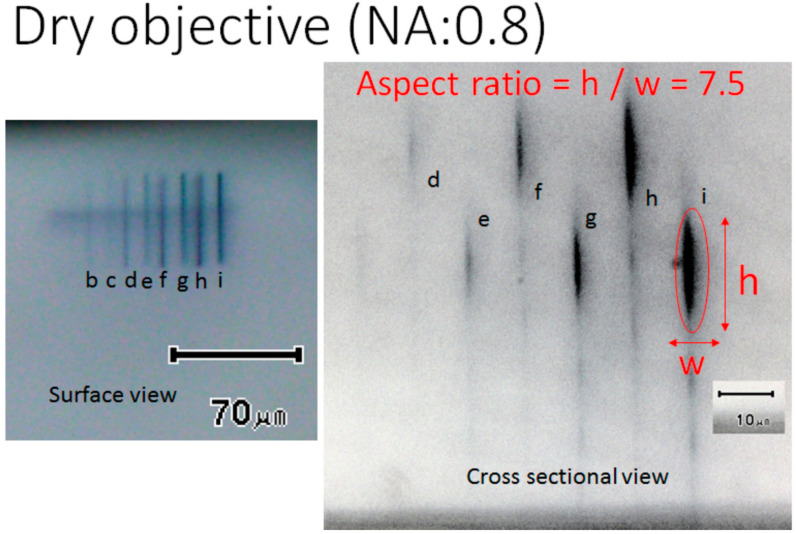
Surface and cross-sectional optical micrographs of photofabricated poly(3,4-ethylendioxythiophene) (PEDOT) line patterns in the transparent Nafion 117 sheet. The photofabrication was carried out by using the dry objective lens (NA: 0.8) and laser focus scanning with a pitch of 0.2 μm, wait of 50 ms, incident laser exposure of a: 0.12, b: 0.19, c: 0.25, d: 0.32, e: 0.38, f: 0.45, g: 0.52, h: 0.61, i: 0.645 mWs, and a repetition rate of 8 MHz. The laser scans were performed nine times, but eight lines were observed from the surface view. (Among them, only five lines could read the dimensions.).

**Figure 3 micromachines-11-01120-f003:**
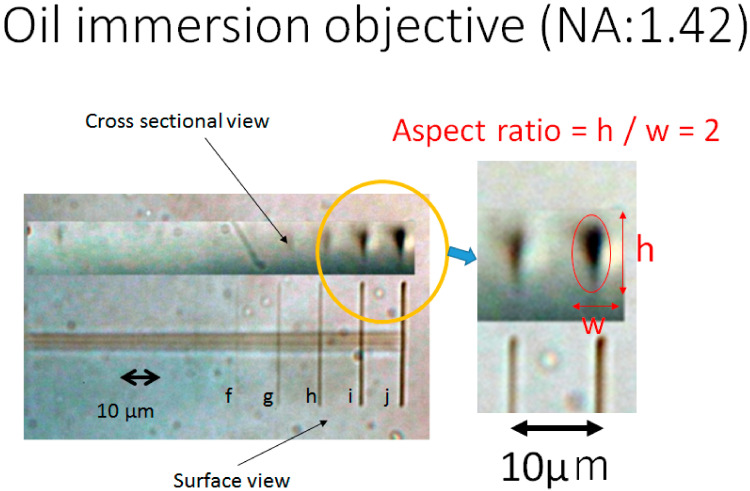
Surface and cross-sectional optical micrographs of photofabricated PEDOT line patterns in transparent Nafion 212 sheet. The photofabrication was carried out by using the oil immersion objective lens (NA: 1.42) and laser focus scanning with a pitch of 0.05 μm, a wait of 100 ms, incident laser exposure of a: 0.01, b: 0.03, c: 0.07, d: 0.11, e: 0.14, f: 0.19, g: 0.24, h: 0.28, i: 0.32, j: 0.375 mWs, and a repetition rate of 8 MHz. The laser scans were performed 10 times, but five lines were observed from the surface view. (Among them, only four lines could read the dimensions.).

**Figure 4 micromachines-11-01120-f004:**
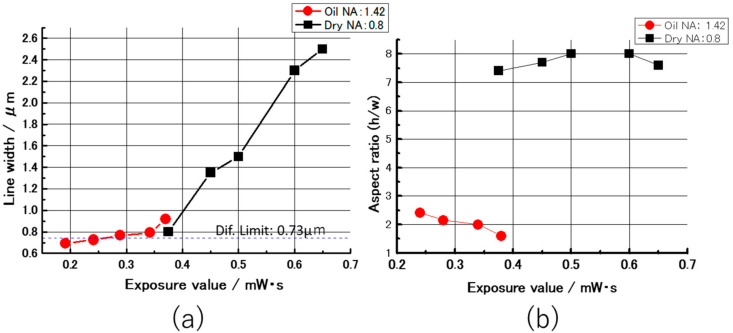
Relationships between the incident laser exposure and the line width of the PEDOT deposition (**a**), and relationships between the incident laser exposure and the aspect ratio (**b**).

**Figure 5 micromachines-11-01120-f005:**
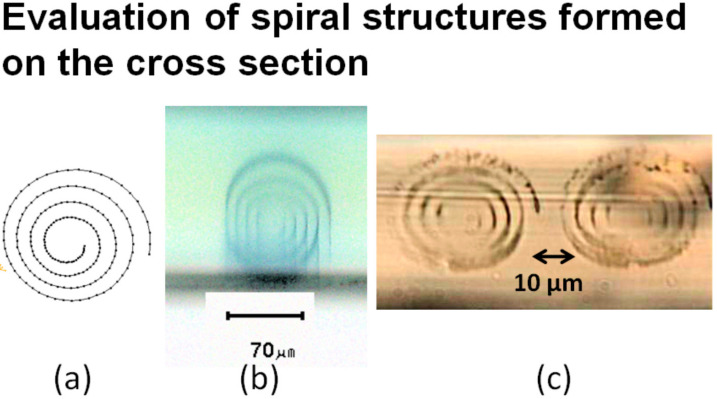
Design of flat spiral pattern (**a**), cross-sectional optical micrograph of the PEDOT pattern photofabricated from the laser focus scanning with the dry objective (**b**), cross-sectional optical micrograph of the PEDOT pattern photofabricated from the laser focus scanning with the oil immersion objective (**c**).

**Figure 6 micromachines-11-01120-f006:**
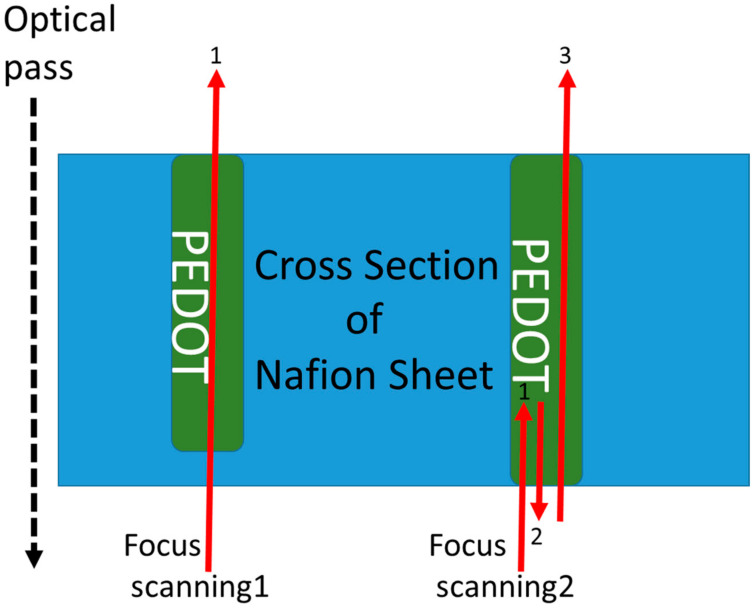
Schematic diagram at the cross section of the Nafion sheet to explain the difference in the procedure of the laser focus scanning in the sheet. Focus scanning 1 and 2 correspond to the previous conductivity evaluation method and the new method, respectively.

**Figure 7 micromachines-11-01120-f007:**
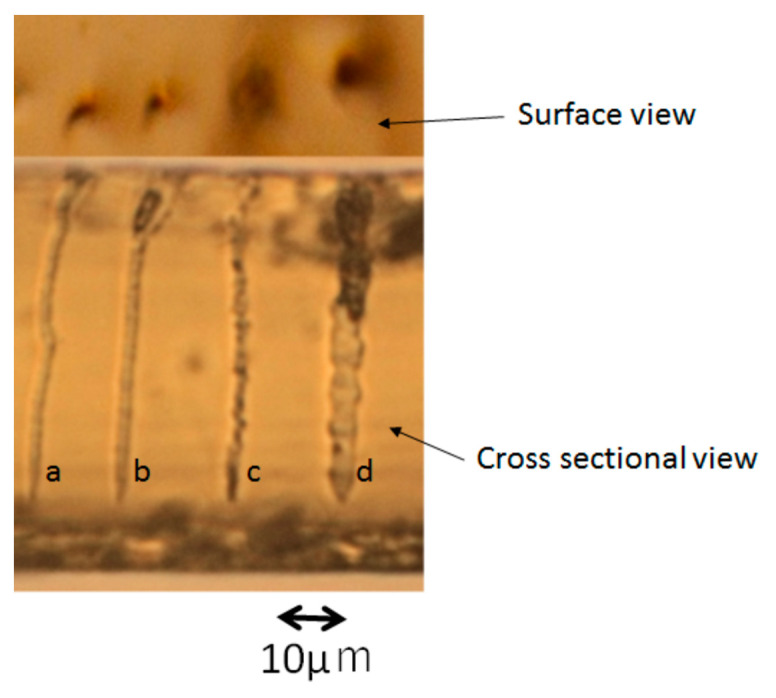
Surface and cross-sectional optical micrographs of photofabricated PEDOT line patterns in transparent Nafion 212 sheet. The photofabrication was carried out by using an oil immersion objective lens (NA: 1.42) and a laser focus scanning with a pitch of 0.05 μm, wait of 100 ms, incident laser exposure of a: 0.015, b: 0.019, c: 0.033, d: 0.041 mWs, and a repetition rate of 8 MHz.

**Figure 8 micromachines-11-01120-f008:**
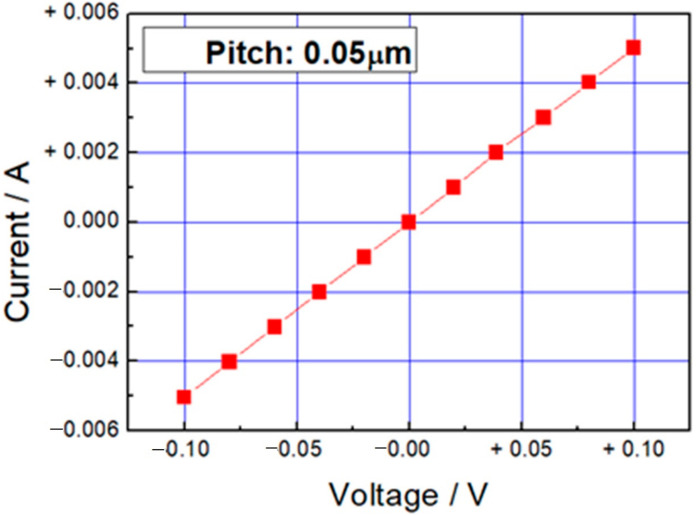
I-V characteristics of the photofabricated PEDOT line pattern in the Nafion 212 sheet (Thickness: 50 μm). The photofabrication was carried out by using an oil immersion objective lens (NA:1.42) and a laser focus scanning with a pitch of 0.05 μm, a wait of 100 ms, incident laser exposure of 0.041 mWs, and a repetition rate of 8 MHz.

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
