# Peer review of "A New 3D Printing System of Poly(3,4-ethylenedioxythiophene) for Realizing a High Electrical Conductivity and Fine Processing Resolution"

_micromachines, 2020, doi:10.3390/mi11121120_

Round 1

Reviewer 1 Report

The manuscript titled “A New 3D Printing System of Poly(3,4- ethylenedioxythiophene) for Realizing a High Electrical Conductivity and Fine Processing Resolution” describes the micro-nano 3D printing of PEDOT polymer. The authors have introduced an oil immersion objective lens into the 3D photofabrication system using a femtosecond pulsed laser as the light source. The authors have claimed the improved electrical properties in 3D PEDOT. The results look promising. However I have few comments on this work:

  1. Why the authors used Nafion sheet for 3D printing of PEDOT.
  2. The 3D printing of PEDOT:PSS using the conducting polymer ink is reported in Yuk, H., Lu, B., Lin, S. et al. 3D printing of conducting polymers. Nat Commun 11, 1604 (2020). https://doi.org/10.1038/s41467-020-15316-7. However, the authors did not cite this article.
  3. Electrical conductivity is 3500 S/m in the 3D PEDOT. What is the reason for this.
  4. There is discontinuity in the spring ring of 3D PEDOT (Fig 5 C). How the authors will fix this.
  5. The spring ring width is not same everywhere. Authors must optimize it. Otherwise, it cannot be considered as reproducible experiment.
  6. The authors have included only one spiral ring of 3D PEDOT. They should include more rings from the experiment.

Reviewer 2 Report

The authors show away to 3D-print PEDOT by using a femtosecond pulsed laser and an objective lense. Hence they achieve a high electrical conductivity.

General remarks:

Improvement of English language is strongly recommended. Examples:

  • line 66 “[PEDOT] is one of the most industrially most successful conductive polymers”
  • line 70 “were carried out using a new system in the this study”. But there are flaws within the whole text, making it sometimes difficult to understand it.

Please have an additional proofreading regarding typos. Examples:

  • Line 123: “shown in Figure.4b” Delete the dot between “Figure” and “4b”.
  • Line 152: a space is missing between “dry objective” and “(b)”
  • Sometimes the space between number and unit is missing

Please give suppliers and article numbers for all materials, chemicals and others like laser, pulse selector etc., used.

How were the images and cross sections of the samples like in Fig 2 and 3 obtained? By which method, which machine, how where cross sections fabricated?

How often were sample production and measurements repeated? In your work just single images and numbers are shown like in Fig 3 and Fig 4. It is unclear to the reader how many times you repeated the production and measurement. So it is unclear if the data points in Fig 4 or 8 and the numbers given throughout the text are obtained from single measurements or are mean values of repeated measurement, and in best case mean values of several samples repeatedly produced. If numbers are mean values (at least after the revision), errors or standard deviation should be given, too.

Figure 2, 3 and 7: Please state in the figures caption that the incident laser exposure is referring to the lines shown from left to right. And please state how many lines were processed, because the ones processed with lower laser exposure can rarely be seen. And what was the increment of laser exposure raise? I assume it was linear, but don’t know.

Remarks regarding special passages in the text:

Line 90: “…are required in order to obtain the line product in this 3D printing system” I’m sorry, but I don’t understand the meaning of this sentence. Could you please rephrase it?

Line 93: “The Nafion sheet was then washed” Using which solvent?

Line 95: How where Au films immobilized in the Nafion?

Line 96: Which equipment was used to measure the conductivity?

Line 167/168: “In particular, the contact is probably hard to form on under the outside surface of the Nafion sheet.” I’m sorry, but I don’t understand the meaning of this sentence. Could you please rephrase it?

Line 182/182: “Because the laser scanning was turned back many times, …” I’m sorry, but I don’t understand the meaning of this sentence. Could you please rephrase it?

Figure 6: This is showing a cross section of the Nafion sheet, isn’t it? If yes, please state that in the figures description.

Figure 7: What is the inset on the lower right of the figure showing? Should this be a length scale? Please make this clearer and explain in the figures caption if needed.

Round 2

Reviewer 1 Report

The authors have modified the manuscript according to the comments. So the manuscript can be accepted for publication.

Reviewer 2 Report

Dear authors,

thank you for the made changes. The article is ok now.